# Influence of Emotional Skills on Attitudes towards Communication: Nursing Students vs. Nurses

**DOI:** 10.3390/ijerph20064798

**Published:** 2023-03-09

**Authors:** María del Carmen Giménez-Espert, Sandra Maldonado, Vicente Prado-Gascó

**Affiliations:** 1Nursing Department, Faculty of Nursing and Chiropody, University of Valencia, Avd/ Menéndez Pelayo, s/n, 46010 Valencia, Spain; 2Nursing Department of the School of Health Sciences, Human Services and Nursing, Lehman College, CUNY, 250 Bedford Park West, New York, NY 10468, USA; 3Social Psychology Department, Faculty of Psychology, University of Valencia, Av. Blasco Ibáñez, 21, 46010 Valencia, Spain

**Keywords:** attitude, communication, cross-sectional, empathy, emotional intelligence, nurses, nursing students

## Abstract

Communication in nursing is essential to the quality of care and patients’ satisfaction, and personal variables such as empathy and emotional intelligence (EI) can improve it; however, no studies have to date analyzed these competencies and their relations among nursing students compared with nurses. The aims of this study are, therefore, to analyze the differences between nursing students and nurses in the means for empathy, EI and attitudes towards communication in order to assess the impact of empathy and EI on nurses’ and nursing students’ attitudes towards communication, and their influence on the behavioral dimension of attitude. A cross-sectional descriptive study was performed on a convenience sample of 961 nursing students and 460 nurses from the Valencian Community, Spain. T-test and hierarchical regression models (HRM) were used. The data was collected in the selected universities in the 2018/2019 academic year. The results showed high levels in all the variables analyzed (i.e., empathy, EI, and attitudes towards communication) in both samples. The HRM results suggested that empathy was a better predictor than EI of the attitudes towards patient communication among both the nursing students and nurses. In the behavioral dimension of the attitude, the cognitive and affective dimensions had greater weight than the emotional component (i.e., empathy and EI). Developing empathy and the cognitive dimension of the attitude in nursing students and nurses could, therefore, help improve EI and attitudes towards communication. These findings are important for developing intervention programs adjusted to real needs.

## 1. Introduction

Nurse-patient communication is interpersonal communication aimed at assessing the patient’s real needs and thereby establishing a therapeutic relationship with him or her [1]. Therapeutic communication is a process of information exchange, an essential competence that nurses need to acquire to understand patients’ needs, in order to plan appropriate care [2]. It is considered one of the most important communication methods and a central element of nursing care for the quality of care and patient satisfaction [3,4,5]. Nurses’ ability to communicate effectively has been associated with patient engagement, and improved health outcomes [6]. In multidisciplinary teams, it is important for healthcare teamwork [7] and fewer adverse patient events [8].

In this context, and for appropriate, safe and quality nursing care [4,5], studying nursing students’ attitudes, in this case towards patient communication, is vital for understanding their beliefs and behaviors, and is a key factor in determining what is relevant in the process of care and healing [9]. Furthermore, the assessment of nursing students’ attitudes towards patient communication is therefore critical for a better understanding of behavior and to ensure adequate communicative competence [1]. The “Theory of Reasoned Action” [10] shows the relationship between the attitudes and behavior of individuals. According to this theory, a change in a person’s attitude can motivate a change in his or her behavior, and as such, studying attitudes determines the probability of an individual performing the behavior in question [11]. An attitude towards communication is made up of three dimensions: cognitive (beliefs and thoughts about communication), affective (positive or negative emotions of a person towards communication) and behavioral (the behavioral disposition, which is a manifestation of the cognitive and affective dimensions) [12]. This aspect is very important for nursing students and nurses, because the assessment of attitudes towards patient communication identifies negative attitudes that can influence the effectiveness of the educational process [13] and the quality of nursing care [14]. Developing favorable attitudes towards communication in nursing students is, therefore, critical [15], and communication is important in improving patient care [16].

Additionally, the literature suggests the existence of personal variables that influence communication with the patient, such as emotional intelligence (EI) and empathy [17,18]. Nurses must understand the needs of their patients, i.e., they must be able to adequately organize their emotions and those of others (EI) and above all, be able to put themselves in the other’s place (empathy) [19], as fundamental aspects for effective communication. Patient safety and nurses’ well-being can be jeopardized if the use of these skills is poor [20]. Nursing education should also teach communication skills, EI and empathy for adequate patient attention [21,22]. These skills are necessary for safe nursing care, teamwork, health care and the management skills essential for decision making and problem solving [23,24]. These aspects are also critical for the delivery of safe and quality nursing care to patients [23]. Communication skills, empathy and EI are fundamental aspects for identifying patients’ needs, increasing nurses’ clinical competence and providing better patient care [25]; however, they are not evaluated during nursing students’ training [21]. Although the literature indicates that the levels of these variables may vary according to the participants’ characteristics and as the training of nursing students progresses, after being exposed to complex real-life situations and human suffering without adequate preparation and training [26,27,28], the explanation may lie in the time constraints of curricula which lead to nurses’ clinical competence being prioritized rather than their emotional skills [2].

According to the literature, the most patient complaints in hospitals are caused by healthcare professionals’ negative attitudes or poor communication skills [29,30]. EI and empathy are an integral part of nurse–patient communication, and can be learned and developed through education [31]. These skills are interrelated in the development of nursing care empathy and EI are positively related to nurses’ job satisfaction [32], students’ mental health and academic performance [33], as well as professional attitude [30]. They also improve the effectiveness of nurses, reducing the incidence of adverse events and job burnout, enhance the nurse–patient relationship and the quality of nursing care overall [34,35]. Training in these skills is, therefore, an important tool for effective communication, EI, empathy and patient care [36], as well as for the well-being of nurses [37]. The adoption of a framework in universities with the fundamentals of nursing care focused on the essential needs of the person to ensure their physical and psychosocial well-being is, therefore, essential in the education of nursing students [38]. These needs are met by developing a positive and trusting relationship with the person being cared for, as well as with his or her family [39]. A positive and trusting nurse–patient relationship based on effective communication is essential for providing high-quality basic care [25]. Once this relationship is established, the nurse can work to meet the patient’s fundamental psychosocial and physical needs. Meeting these needs is, in turn, mediated by nurses’ emotional skills such as communication, empathy, and EI [37,40,41]. Another key element is the care context, including health policies that may hinder or facilitate high-quality basic care [42]. In addition, the importance of the global nursing workforce in the sustainability of health systems has been demonstrated in the context of the COVID-19 pandemic [43]. The training of nurses and nursing students as future professionals in these aspects is, therefore, essential for improving their well-being and job satisfaction [37]. These are crucial issues, since 43% of nurses in Europe consider leaving their job within three years [44].

Despite its importance, the literature on the effect of EI and empathy and the attitudes on communication behavior in nursing students is scarce. Some studies in the nursing context have only used regression models in nurses [45,46]. Other studies have used regression models and fsQCA models to analyze these variables in nurses [47]. Several research studies of nursing students have showed that patient communication is highly related to empathy [26,48] and is associated with EI [49,50]; however, no studies have used regression models in the same research to analyze these variables in nursing students and nurses. The main aims of this study are, therefore, (1) to explore the differences between nursing students and nurses in the means of the variables analyzed (i.e., empathy, EI and attitudes towards communication); and (2) to assess the relationship between empathy and emotional intelligence (EI) as the predictors of nursing students’ attitudes towards communication, and their influence on communication-related behavior using regression, making a comparison with the results in nurses [47]. On the basis of the previous literature, the following hypotheses delimited this study:

**H1:** 
*All analyzed variables (empathy, EI and attitudes towards communication) will show statistically significant differences in their means in nursing students and nurses.*


**H2:** 
*Empathy and EI will have a negative effect on the affective dimension and a positive effect on the cognitive and behavior dimension of attitudes towards communication in nursing students and nurses.*


It is also to be hoped that this will help healthcare managers and academics to establish academic plans for nursing students and nurses to develop these emotional skills and improve their attitudes towards communication, in order to ultimately provide quality nursing care.

## 2. Materials and Methods

### 2.1. Design and Participants

A cross-sectional descriptive study was conducted in a convenience sample of 961 nursing students, recruited in five universities in the Valencian Community, Spain. The sample size was determined according to data available from the Spanish Ministry of Universities, of 5062 nursing students enrolled in the Valencian Community in the 2018/2019 academic year, with a margin of error of 2.85% and a confidence level of 95% [51]. A cross-sectional study is a type of observational study design that involves analyzing data from a population at a specific point in time, with no prospective or retrospective follow-up. The subjects are selected from a population available for the study question, showing the “point in time” situation of a sample of subjects [52,53]. Convenience sampling is a non-probability sampling method, in which the sample is selected according to a subjective judgment of the researcher, based on the subjects’ availability and willingness to participate [53,54]. The inclusion criteria were (i) voluntarily agreeing to participate and (ii) signing the informed consent. The exclusion criteria were not having been diagnosed with a severe physical or psychiatric illness preventing the student’s presence during the application of the instrument. The data were collected by researchers at the selected universities in the 2018/2019 academic year, using an instrument that included sociodemographic variables (such as sex, age, year of degree course and employment situation of the respondents), attitudes towards communication (ACO) adapted to nursing students [28], the Trait Meta-Mood Scale (TMMS24) [55] and the Jefferson Scale of Empathy (JSE) for nursing students adapted and validated in Spain by Giménez-Espert et al. [55]. Approximately 20 min. were required to complete the instrument, and it was completed in the classroom by the students. All the participants were informed of the aims of the study, the confidentiality of the data, and that non-participation would not have any consequences for their academic development.

The comparison sample was a convenience sample of 460 nurses enrolled from 6 hospitals in the Valencian Community, Spain [47]. The sample size was determined according to data available from the Spanish National Statistics Institute, with a total of 26,565 nurses working in the Valencian Community in 2015, with a margin of error of 4.53% and a confidence level of 95% [51]. The data collection from the nurses was conducted by researchers in September 2015–February 2016. The instrument contained sociodemographic variables, attitudes towards communication of nurses (ACO) [56], the Trait Meta-Mood Scale (TMMS24) [57], and the Jefferson Scale Nursing Empathy (JSNE) [58,59]. Approximately 35 min. were required to complete the instrument. The researchers visited the hospitals involved in the study. All the participants received detailed information about the aims and procedures, and were informed regarding confidentiality, with an emphasis on the anonymity of the data collected and the non-discrimination of participants. Once the instruments had been submitted, reminders were sent via e-mail after 2 weeks and after 3–4 weeks. The completed instruments were subsequently collected from the various boxes placed in the different services for this purpose [47].

This study complied with the basic principles of the Helsinki Declaration (World Medical Association, 2013). It was authorized by the Human Research Ethics Committee of the University of Valencia (H1529396558647).

### 2.2. Measures

Only information regarding the student sample is presented below. The information related to the nurses can be consulted in Giménez-Espert and Prado-Gascó [47].

- Attitudes towards communication with the patient (ACO) for nursing students [28] (intellectual property UV-MET-201917R registered at the University of Valencia on 8 April 2019). The instrument consisted of 25 items grouped in three dimensions: affective related to patients’ admission, procedure and discharge that create anxiety in nurses, which were written in the negative or reverse to avoid the phenomenon of acquiescence [60] (i.e., 12 items, Cronbach’s α = 0.95, e.g., “I’m nervous when I give time to the patient and/or family to ask questions and express their concerns”); behavioral, related to information on admission, obtaining informed consent and information on discharge (i.e., 9 items, Cronbach’s α = 0.92, e.g., “I usually check that the patient and / or family has understood the discharge information”); and cognitive, referring to the importance of the information that can help in recovery, on discharge care and finally a collaboration with other members of the healthcare team (i.e., 4 items, Cronbach’s α = 0.85, e.g., “I need to work with other healthcare team members to provide information to enable the continuity of care.”). A five-point Likert scale was used, ranging from 1 = strongly disagree to 5 = strongly agree.- Empathy was measured using the Jefferson Scale of Empathy (JSE) for nursing students, adapted and validated in Spain by Giménez-Espert et al. [55]. This is a 19 item scale composed of three factors and adequate psychometric properties; perspective taking (i.e., 10 items; α Cronbach = 0.83, e.g., “Patients value a nurse’s understanding of their feelings that is therapeutic in its own right”), which is the central cognitive ingredient of empathy; compassionate care (i.e., 7 items; α Cronbach = 0.81, e.g., “Asking patients about what is happening in their personal lives is not helpful in understanding their physical complaints”), which aims to understand the patient’s experiences, and feelings; thinking like the patient (i.e., 2 items; α Cronbach = 0.84, e.g., “Nurses understanding of their patients feelings and the feelings of their patients families does not influence nursing care”), which is understood as putting oneself in a patient’s place [55]. The two last dimensions were written in the negative or reverse to avoid the phenomenon of acquiescence [60]. A five-point Likert scale was used, ranging from 1 = strongly disagree to 5 = strongly agree.- Emotional intelligence was measured using the TMMS24. The Trait Emotional Meta-Mood Scale was adapted and validated for Spanish nursing students by Giménez-Espert et al. [55]. It is a 24 item instrument grouped into three dimensions: emotional attention refers to the person’s attention to his/her feelings and emotions (i.e., 8 items; α Cronbach = 0.86; e.g., “I pay a lot of attention to my feelings”); emotional clarity, i.e., the perceived ability to understand and discriminate between feelings (i.e., 8 items; α Cronbach = 0.89; e.g., “Sometimes I can tell what my feelings are”); and emotional repair, i.e., the perceived ability to regulate mood states, repairing negative emotional experiences and prolonging positive ones (i.e., 8 items; α Cronbach = 0.86; e.g., “I have a lot of energy when I am happy”). A five-point Likert scale was used, ranging from 1 = strongly disagree to 5 = strongly agree.

### 2.3. Data Analysis

The SPSS 23.0 package (IBM Corporation, NY, USA) was used for the statistical analysis, and the first descriptive analyses of the participants were measured. Afterwards, t- tests were carried out to determine the difference in the means for nursing students and nurses in the variables (i.e., empathy, EI and attitudes towards communication). In addition, hierarchical regression models (HRM) were used in order to assess the predictive capacity of empathy and EI on the affective and cognitive dimensions of attitudes towards communication among nursing students, and the effect of empathy, EI, affective and cognitive dimensions of attitudes towards communication on behavioral dimensions. In the hierarchical regression models, two models with two steps (step 1: empathy; step 2: EI) and one model with three steps (step 1: empathy; step 2: EI; step 3: affective and cognitive dimensions) were calculated, considering each dimension of attitudes towards communication.

## 3. Results

### 3.1. Sample Characteristics

The study sample included 961 nursing students ranging in age from 17 to 55 years, with a mean age of 21.58 (SD = 5.14). A total of 86% (814) were women and 14% (132) were men. The percentages of the participants as regards each year of their degree course were 32% (295) in their first year, 26.20% (242) in their second year, 22.50% (208) in their third year, and 19.30% (178) in their fourth year. Finally, regarding the employment situation of the nursing students, 11.70% (103) were temporarily employed, 80.30% (704) were not working and 8% (70) had a permanent position.

### 3.2. Descriptive Statistics and T Test

First, the results showed high levels in all the variables analyzed in both samples. Table 1 shows the difference between the nursing students and nurses for the means of the variables analyzed. For the ACO, JSE and TMMS24, statistically significant differences were observed between the nurses and the nursing students in the thinking like the patient dimension of the JSE scale. The nurses showed slightly lower scores for thinking like the patient than the nursing students (Table 1). In this case, as explained above, the dimension of thinking like the patient of the empathy was written in reverse, so that higher levels in this dimension indicate lower empathic orientation in nursing students.

However, when we considered EI in isolation, significant differences were found for emotional attention, clarity and repair on the TMMS24 scale. The nurses showed slightly lower scores for emotional attention than the nursing students, while the nurses showed higher scores for emotional clarity and repair than the nursing students. Finally, as regards the ACO, significant differences between the nurses and nursing students were observed in the cognitive and behavioral dimensions. The nursing students had higher mean scores in the cognitive and behavioral dimensions than the nurses (Table 1).

### 3.3. Hierarchical Regression Model (HRM)

For the nursing students, two steps were established in the model for predicting the affective dimension of ACO (R^2^_adjusted_ = 0.06, *p* ≤ 0.001). First, all the dimensions of JSE were entered (step 1) (R^2^ = 0.54, *p* ≤ 0.001). The TMMS24 components were then included (step 2) (R^2^ = 0.01; *p* ≤ 0.001). When each dimension was considered, only compassionate care (affective: β = 0.09; *p* ≤ 0.01) and thinking like the patient (affective: β = 0.08; *p* ≤ 0.05) were significant positive predictors of affective dimensions, and perspective-taking (affective: β = −0.15; *p* ≤ 0.001) and emotional repair (affective: β = −0.08; *p* ≤ 0.05) were significant negative predictors of affective dimensions (Table 2).

Two steps were established in the model for the prediction of the cognitive dimension of ACO among the nursing students (R^2^_adjusted_ = 0.18, *p* ≤ 0.001). First, the JSE components of empathy were entered (step 1) (R^2^ = 0.17, *p* ≤ 0.001). Then, all the trait components of EI were included (step 2) (R^2^ = 0.01, *p* ≤ 0.01). The perspective-taking of empathy (cognitive: β = 0.35; *p* ≤ 0.001) was a significant positive predictor, and the compassionate care of empathy (cognitive: β = −0.16; *p* ≤ 0.001), emotional attention (cognitive: β = −0.70; *p* ≤ 0.05) and repair (cognitive: β = −0.70; *p* ≤ 0.05), were significant negative predictors of cognitive dimension.

Three steps were established in the model for the prediction of the behavioral dimension of ACO (R^2^_adjusted_ = 0.73, *p* ≤ 0.001). First, all the dimensions of the instrument JSE were entered (step 1) (R^2^ = 0.15, *p* ≤ 0.001). The TMMS24 dimensions were then included (step 2) (R^2^ = 0.20, *p* ≤ 0.001), and the affective and cognitive dimensions were entered (step 3) (R^2^ = 0.57, *p* ≤ 0.001), according to the theoretical foundation. In the third step of prediction, the behavioral dimension of perspective-taking (β = 0.04; *p* ≤ 0.05) was a significant and positive predictor, while the thinking like the patient aspect of empathy (β =−0.05; *p* ≤ 0.01) was a significant negative predictor. The emotional clarity of EI (β = 0.05; *p* ≤ 0.01) was a significant and positive predictor in the behavioral dimension. In the third step, attitudinal variables were more important than empathy or EI, with the best predictors being the cognitive dimension (β = 0.80; *p* ≤ 0.001) and the affective dimension (β =−0.09; *p* ≤ 0.001).

EI and empathy accounted for 6% of the variance in the affective dimension, 18% of the variance in the cognitive dimension and 20% of the variance in the behavioral dimensions. Empathy is a better predictor than EI in all cases, and especially the perspective-taking dimension in the cognitive and behavioral dimensions. In addition, in the prediction of the behavioral dimension, the attitude components (i.e., affective and cognitive dimensions) (R^2^ = 0.57; *p* ≤ 0.001) accounted for a greater degree of variance as emotional components. In general, empathy (perspective taking) and EI (emotional clarity and repair) predicted the ACO (affective and cognitive dimensions). These results for nurses were presented in Giménez-Espert and Prado-Gascó [47].

## 4. Discussion

This research analyzed the differences between nursing students and nurses for the means of the variables measured (i.e., empathy, EI and attitudes towards communication), as well as the impact of empathy and EI on nursing students’ attitudes towards patient communication and their influence on behavior. This study is the first in nursing students that attempts to identify the relationships between emotional and attitudes towards communication; therefore, the results obtained may be useful for improving nursing education and the quality of nursing care [23,34,35].

In general, as was expected and postulated in the first hypothesis, statistically significant differences were observed for almost all the variables considered (with the exception of perspective taking, compassionate care and the affective dimension). In general, higher levels were observed for the students than the nurses, except in the case of the thinking like the patient dimension of empathy written in negative, and emotional clarity and repair, where this situation is reversed. These results may be explained by the fact that as clinical experience in real and complex contexts increases, there is a decline in empathy in nursing students [61,62]. High levels of emotional attention with low levels of emotional clarity and repair are related to inadequate coping strategies (i.e., more passive and less active), and psychological problems [63,64], as an adaptive response to new responsibilities and an increasing workload [23]. For example, attention is focused on tasks and technology, to the detriment of relations with the patient [65]. Nursing students distance themselves from a patient’s suffering as their clinical experience increases [4,66], which leads to lower levels of these emotional variables affecting patient safety [20]. This is because their training has been more focused on clinical skills, and they feel poorly trained to deal with these situations [21,22].

According to the second hypothesis, empathy and EI will have a negative effect on the affective dimension and a positive effect on the cognitive and behavior dimension of attitudes towards communication in nursing students and nurses. The HRM results suggested that hypothesis H2 can be partially accepted. The relationships observed were in the direction hypothesized, although not all the dimensions were equally significant. Empathy and EI had a negative effect on the affective dimension and a positive effect on the cognitive and behavior dimension of attitudes towards communication in nursing students and nurses. Only the perspective-taking in empathy and the emotional repair dimension of the EI were significant negative predictors in the affective dimension, and compassionate care and thinking like the patient in empathy, was a significant positive predictor for the affective dimension in the nursing students. Meanwhile, in the nurses, the perspective-taking in empathy and emotional clarity dimension of EI were significant negative predictors in the affective dimension of attitudes towards communication. The perspective-taking in empathy was a significant positive predictor of the cognitive dimension in the nursing students and in the nurses. The compassionate care of empathy and the emotional attention and repair of EI were significant negative predictors of the cognitive dimension in nursing students. The emotional clarity was a significant positive predictor of the cognitive dimension in nurses. In the behavioral dimension, perspective-taking was a significant and positive predictor, while thinking like the patient of empathy was a significant negative predictor, in the nursing students and nurses. In the nurses, emotional repair, and in the nursing students, emotional clarity, were significant and positive predictors. In addition, attitudinal variables were more important than empathy or EI in the nursing students and nurses, with the best predictors being the cognitive dimension (a significant positive predictor) and the affective dimension (a significant negative predictor).

Likewise, it seems that emotional variables were in general better predictors for the nurses than for the students, especially the perspective-taking dimension of empathy in the cognitive, affective and behavioral dimensions in both the samples of nursing students and nurses. In general, empathy (perspective taking) and EI (emotional clarity in nurses, and emotional repair in the nursing students) predicted their attitude towards communication (affective and cognitive dimensions). This could be because the perspective-taking dimension of empathy is the central cognitive axis of empathy, emotional clarity and emotional repair (where high levels indicate adequate emotional adjustment) promoting favorable attitudes towards communication in order to better understand the patients [40,67]. In the behavioral dimension of attitude towards communication, the emotional component (EI and empathy) is a weaker predictor than the affective and cognitive dimensions, and the inclusion of the attitudinal component (affective and cognitive) significantly increased the prediction to 57% of the variance in the behavioral dimension of the attitude [47]. These findings can be explained by the fact that the three components of attitude are related, so that feelings (the affective dimension) are based on knowledge, in this case about communication (the cognitive dimension), and communicative behavior (the behavioral dimension) is based on feelings and knowledge about communication [21,68]. The cognitive and affective dimensions were better predictors of the behavioral dimension, as shown in the literature [13,28]. Moreover, empathy may be a predictor of EI [37,40]. High levels of empathy provide an understanding of internal feelings and those of others, distinguishing emotions as well as enabling listening [69], and more harmonious interpersonal relationships [70,71], which are key elements of EI [68]. In fact, some authors include empathy as a part of EI [37,72,73]; therefore, according to the results, in nursing it seems more advisable to focus first on forming the cognitive and affective aspects of attitudes towards communication through training in emotional competency, especially empathy, for both nurses and nursing students in order to promote the behavioral aspect of attitudes towards communication with patients [74].

Despite the interest of this study, one of the major limitations is its convenience sampling. It is difficult to generalize the results, and the cross-sectional nature of the study type meant it was impossible to establish causal relationships. In future research, it would be interesting to extend the study sample to other Spanish-speaking countries, and to establish a longitudinal design that would enable causal relationships to be established, and to include other variables that may influence the results, such as age and gender [35]. Another limitation is related to the use of self-reporting, which can introduce social-desirability bias [60]. It would be useful to use another type of instrument completed by others, and/or one with external objective measures.

As for the practical implications of this research, given that EI and empathy are an integral part of nurse-patient communication, and can be learned and developed through education [75], these results could help managers in academic and healthcare institutions to highlight the importance of empathy in improving attitudes towards communication with patients by both nursing students and nurses to promote their well-being and provide high quality nursing care [23,34,35]. This is a key area of nursing education, as resolving a wide variety of complex situations with patients requires emotional skills and positive attitudes towards communication [76]. Including education about emotional skills and communication in the nursing curricula, and in continuing education strategies for nurses is, therefore, a necessity [77]. Role-playing exercises [78] as well as clinical simulations [41] could be useful. Nurses could simulate real situations to feel the patient’s physical state and emotional experience, which could improve nurses’ EI and empathy. The educators of nurses could assess EI, empathy, and attitudes towards communication among nursing students on their admission to training in order to develop an individual support plan for improvement [16], as well as in nurses, paying more attention to psychological counseling in their work environment, especially in the wake of the COVID-19 pandemic, and through regular continuing education courses to improve these skills [37].

## 5. Conclusions

This research can be considered an initial approach to the study of EI and empathy as predictors of nursing students’ attitudes towards communication compared with a sample of nurses. Developing empathy and the cognitive dimension of attitudes towards communication in nursing students and nurses could help to improve EI and attitudes towards communication with patients and their families. The findings of the study produce scientific evidence for further investigation to identify and develop intervention programs adjusted to real needs based on individual support plans, aimed at improving the education of nursing students and involving regular continuing education courses for nurses in order to ensure their well-being and improve the quality of patient care.

## Figures and Tables

**Table 1 ijerph-20-04798-t001:** Differences in the means of the variables analyzed between nursing students and nurses.

	**Nursing Students**		**Nurses**			
	M	SD	M	SD	*t*	*p*
PT	4.52	0.45	4.52	0.57	−0.21	0.834
CC	1.88	0.73	1.88	0.90	−0.01	0.991
TP	2.35	1.03	2.05	1.04	5.25	0.000
EA	3.63	0.71	3.58	0.77	1.08	0.028
EC	3.54	0.75	3.84	0.69	−0.71	0.000
ER	3.70	0.73	3.82	0.77	−3.01	0.003
Affective	1.65	0.67	1.60	0.89	0.98	0.328
Cognitive	4.73	0.63	4.49	0.82	5.53	0.000
Behavioral	4.58	0.68	4.20	0.81	8.53	0.000

Note: PT: perspective taking; CC: compassionate care; TP: thinking like the patient; EA: emotional attention; EC: emotional clarity; ER: emotional repair.

**Table 2 ijerph-20-04798-t002:** Hierarchical regressions for the dimensions of TMMS24 and JSE and on ACO dimensions (Affective, Cognitive and Behavioral).

	Nursing Students								Nurses			
Variable	Affective		Cognitive		Behavioral			Affective		Cognitive		Behavioral
Predictors	∆R^2^	β	∆R^2^	β	∆R^2^	β	∆R^2^	β	∆R^2^	β	∆R^2^	β
Step 1	0.54 ***		0.17 ***		0.15 ***		0.18 ***		0.21 ***		0.30 ***	
Perspective-taking		−0.17 ***		0.35 ***		0.34 ***		−0.35 ***		0.40 ***		0.49 ***
Compassionate care		0.07 *		−0.16 ***		−0.10 *		0.09		−0.10		−0.03
Thinking like the patient		0.08 **		0.05		−0.03		0.06		−0.02		−0.13 *
Step 2	0.01 ***		0.01 **		0.20 ***		0.03 **		0.02 *		0.03 **	
Perspective-taking		−0.15 ***		0.35 ***		0.31 ***		−0.34 ***		0.38 ***		0.42 ***
Compassionate care		0.09 **		−0.16 ***		−0.11 ***		0.08		−0.09		−0.02
Thinking like the patient		0.08 *		0.04		−0.03		0.05		−0.02		−0.11 *
Emotional attention		0.05		−0.70 *		−0.04		0.05		−0.04		−0.03
Emotional clarity		−0.05		−0.00		0.05		−0.20 **		0.17 **		0.17 **
Emotional repair		−0.08 *		−0. 70 *		0.10 **		0.10		−0.05		0.07
Step 3	-		-		0.57 ***				-		0.31 ***	
Perspective-taking		-		-		0.04 *		-		-		0.14 **
Compassionate care		-		-		0.02		-		-		0.06
Thinking like the patient		-		-		−0.05 **		-		-		−0.10 **
Emotional attention		-		-		0.01		-		-		0.01
Emotional clarity		-		-		0.05 **		-		-		0.02
Emotional repair		-		-		0.02		-		-		0.11 **
Affective		-		-		−0.09 ***		-		-		−0.19 ***
Cognitive		-		-		0.80 ***		-		-		0.52 ***
Total R^2^_adjusted_	0.06 ***		0.18 ***		0.73 ***		0.19 ***		0.22 ***		0.64 ***	

Note: * *p* ≤ 0.05; ** *p*≤ 0.01; *** *p* ≤ 0.001;—were not calculated according to the theoretical model; results for nurses [47].

## Data Availability

The data presented in this study are available on request from the corresponding author.

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
