# Peer review of "Influence of Emotional Skills on Attitudes towards Communication: Nursing Students vs. Nurses"

_ijerph, 2023, doi:10.3390/ijerph20064798_

Round 1
Reviewer 1 Report
Dear Authors,
In my opinion the aim should be formulated in clearer way, you should also write the hypothesis.
The emotional skills and communication with patients is nearly as important as treatment procedures, so I think you should propose in the article some practical suggestion, guidelines.
Author Response
Dear reviewer,
We would like to thank you for your time and effort in revising our paper. We really appreciate the suggestions made and we believe the document has greatly benefited from it.
In general, abstract, introduction, materials and methods, results, discussion (limitations) and conclusions have been modified. More references have been included to cover the relevant literature. References have been reviewed, and some of them have been adapted in order to comply with the journal´s requirements. The article has undergone a new language revision, certificate of which is attached. All changes have been included in the article in red.
Response to reviewer 1
We appreciate your concern about the aims and the hypotheses of the study. The authors, in accordance with the reviewer's suggestion, have tried to formulate the aims in clearer way and write the hypothesis.
Page 3 line 114-126
“The main aims of this study are therefore (1) to explore the differences between nursing students and nurses in the means of the variables analyzed (empathy, EI and attitudes towards communication); and (2) to assess the relationship between empathy and emotional intelligence (EI) as predictors of nursing students' attitudes towards communication, and their influence on communication-related behavior using regression, making a comparison with the results in nurses [49]. On the basis of the previous literature, the following hypotheses delimited this study:
- H1: All analyzed variables (empathy, EI and attitudes towards communication) will show statistically significant differences in their means in nursing students and nurses.
- H2: Empathy and EI will have a negative effect on the affective dimension and a positive effect on the cognitive and behavior dimension of attitudes towards communication in nursing students and nurses.”
We really appreciate the suggestions made regard to propose in the article some practical suggestion, guidelines. In accordance with your comments, the discussion section has been included the practical implications of this research.
Page 10 and 11, line 395-411
“As for the practical implications of this research, given that EI and empathy are an integral part of nurse-patient communication, and can be learned and developed through education [78], these results could help managers in academic and healthcare institutions to highlight the importance of empathy in improving attitudes towards communication with patients by both nursing students and nurses to promote their well-being and provide high quality nursing care [23, 35,36]. It is a key area of nursing education, as resolving a wide variety of complex situations with patients requires emotional skills and positive attitudes towards communication [79]. Including education about emotional skills and communication in the nursing curricula, and in continuing education strategies for nurses is therefore a necessity [80]. Role-playing exercises [81] as well as clinical simulation [43] could be useful. Nurses could simulate real situations to feel the patient's physical state and emotional experience, which could improve nurses' EI and empathy. Educators of nurses could assess EI, empathy, and attitudes towards communication among nursing students on admission to training in order to develop an individual support plan for improvement [16], as well as in nurses, paying more attention to psychological counseling in their work environment, especially the wake of the COVID19 pandemic, and through regular continuing education courses to improve these skills [38]”.
"Please see the attachment."

Reviewer 2 Report
Dear Authors
First of all, I would like to congratulate you on your work. It is a topic of great interest, taking into account the importance of those non-technical competencies that we health professionals have to acquire.
One of the aspects that should be reviewed is the objective of the study. It is recommended that the objective be clearly stated.
In fact, it is in the section on Materials and Methods where this objective seems to be found.
In this same section you refer to an article to learn about the recruitment of the participating nurses. I would appreciate it if all the information in this regard could be included in the article so that the reader could understand and read the research carried out in a comprehensive manner.
The Jefferson Scale of Empathy does not indicate whether it has been validated and translated into Spanish.
Results:
185." Finally, in terms of their employment situation, 11.7% (103) weretemporarily employed, 80.3% (704) were not working and 8% (70) had a permanent position. "Do you mean nursing students?
The results are difficult to read due to the large number of data and their repetition in the tables. Perhaps it could be clarified.
The discussion lacks comparison with similar studies.
Author Response
Dear reviewer,
We would like to thank you for your time and effort in revising our paper. We really appreciate the suggestions made and we believe the document has greatly benefited from it.
In general, abstract, introduction, materials and methods, results, discussion (limitations) and conclusions have been modified. More references have been included to cover the relevant literature. References have been reviewed, and some of them have been adapted in order to comply with the journal´s requirements. The article has undergone a new language revision, certificate of which is attached. All changes have been included in the article in red.
Response to reviewer 2
We appreciate your concern about the objectives. The authors, in accordance with the reviewer's suggestion, have tried to formulate the objectives in clearer way and write the hypothesis.
Page 3 line 114-126
“The main aims of this study are therefore (1) to explore the differences between nursing students and nurses in the means of the variables analyzed (empathy, EI and attitudes towards communication); and (2) to assess the relationship between empathy and emotional intelligence (EI) as predictors of nursing students' attitudes towards communication, and their influence on communication-related behavior using regression, making a comparison with the results in nurses [49]. On the basis of the previous literature, the following hypotheses delimited this study:
- H1: All analyzed variables (empathy, EI and attitudes towards communication) will show statistically significant differences in their means in nursing students and nurses.
- H2: Empathy and EI will have a negative effect on the affective dimension and a positive effect on the cognitive and behavior dimension of attitudes towards communication in nursing students and nurses.”
We really appreciate the suggestions made about the recruitment of the participating nurses. All relevant information has been included in the article so that the reader can comprehensively understand and read the research carried out.
Page 3-4 line 131-171
“2. Materials and Methods
2.1 Design and participants
A cross-sectional descriptive study was conducted in a convenience sample of 961 nursing students, recruited in five universities in the Valencian Community, Spain. The sample size was determined according to data available from the Spanish Ministry of Universities, of 5,062 nursing students enrolled in the Valencian Community in the 2018/2019 academic year, with a margin of error of 2.85% and a confidence level of 95% [53]. A cross-sectional study is a type of observational study design that involves analyzing data from a population at a specific point in time, with no prospective or retrospective follow-up. Subjects are selected from a population available for the study question, showing the "point in time" situation of a sample of subjects [54,55]. Convenience sampling is a non-probability sampling method, in which the sample is selected according to a subjective judgment of the researcher, based on the subjects' availability and willingness to participate [55,56]. The inclusion criteria were (i) voluntarily agreeing to participate and (ii) signing the informed consent. Exclusion criteria were not having been diagnosed with a severe physical or psychiatric illness preventing the student’s presence during the application of the instrument. The data were collected by researchers at the selected universities in the 2018/2019 academic year, using an instrument that included sociodemographic variables (such as sex, age, year of degree course and employment situation of the respondents), attitudes towards communication (ACO) adapted to nursing students [28], the Trait Meta Mood Scale (TMMS24) [57] and the Jefferson Scale of Empathy (JSE) for nursing students adapted and validated in Spain by Giménez – Espert et al. [57]. Approximately 20 minutes were required to complete the instrument, and it was completed in the classroom by the students. All the participants were informed of the aims of the study, the confidentiality of the data, and that non‐participation would not have any consequences for their academic development.
The comparison sample was a convenience sample of 460 nurses enrolled from 6 hospitals in the Valencian Community, Spain [49]. The sample size was determined according to data available from the Spanish National Statistics Institute, with a total of 26,565 nurses working in the Valencian Community in 2015, with a margin of error of 4.53% and a confidence level of 95% [53]. The data collection from the nurses was conducted by researchers in September 2015–February 2016. The instrument contained sociodemographic variables, attitudes towards communication of nurses (ACO) [58], Trait Meta-Mood Scale (TMMS24) [59], and Jefferson Scale Nursing Empathy (JSNE) [60,61]. Approximately 35 minutes were required to complete the instrument. The researchers visited the hospitals involved in the study. All the participants received detailed information about the aims and procedures, and were informed regarding confidentiality, with an emphasis on the anonymity of the data collected and non-discrimination of participants. Once the instruments had been submitted, reminders were sent via e-mail after 2 weeks and after 3-4 weeks. The completed instruments were subsequently collected from the various boxes placed in the different services for this purpose [49]”.
We appreciate your concern, about the Jefferson Scale of Empathy. The authors have indicated it has been validated and translated into Spanish.
Page 5 line 193-205
“Empathy was measured using the Jefferson Scale of Empathy (JSE) for nursing students, adapted and validated in Spain by Giménez – Espert et al. [57]. This is a 19-item scale composed of three factors and adequate psychometric properties; perspective taking (10 items; α Cronbach = 0.83, e.g. “Patients value a nurse’s understanding of their feelings that is therapeutic in its own right”), which is the central cognitive ingredient of empathy; compassionate care (7 items; α Cronbach = 0.81, e.g. “Asking patients about what is happening in their personal lives is not helpful in understanding their physical complaints”), which aims to understand the patient's experiences, feelings; thinking like the patient (2 items; α Cronbach = 0.84, e.g., “Nurses understanding of their patients feelings and the feelings of their patients families does not influence nursing care”), understood as putting oneself in the patient's place [57]. The two last dimensions were written in the negative or reverse to avoid the phenomenon of acquiescence [62]. A five-point Likert scale was used, ranging from 1= strongly disagree to 5= strongly agree”.
We really appreciate the suggestions made about the results section. The authors have clarified information on the employment status of nursing students. As well as reorganizing and eliminating redundant information from the section to facilitate understanding.
Page 5-8, line 230-310
“3. Results
3.1 Sample characteristics
The study sample included 961 nursing students ranging in age from 17 to 55 years, with a mean age of 21.58 (SD=5.14). 86% (814) were women and 14% (132) were men. The percentages of participants as regards each year of their degree course were 32% (295) in the first year, 26.2% (242) in the second year, 22.5% (208) in the third year, and 19.30% (178) in the fourth year. Finally, regarding the employment situation of the nursing students, 11.7% (103) were temporarily employed, 80.3% (704) were not working and 8% (70) had a permanent position.
3.2 Descriptive statistics and t test
First, the results showed high levels in all the variables analyzed in both samples. Table 1 shows the difference between the nursing students and nurses for the means of the variables analyzed. In for the ACO, JSE and TMMS24 statistically significant differences were observed between the nurses and the nursing students in the thinking like the patient dimension of the JSE scale. The nurses showed slightly lower scores for thinking like the patient than the nursing students (Table 1). In the case, as explained above, the dimension thinking like the patient of the empathy is written in reverse, so that higher levels in this dimension indicate lower empathic orientation in nursing students.
However, when we considered EI in isolation, significant differences were found for emotional attention, clarity and repair on the TMMS24 scale. The nurses showed slightly lower scores for emotional attention than the nursing students, while the nurses showed higher scores for emotional clarity and repair than the nursing students. Finally, as regards ACO, significant differences between the nurses and nursing students were observed in the cognitive and behavioral dimensions. The nursing students had higher mean scores in the cognitive and behavioral dimensions than the nurses (Table 1).
Table 1. Differences in the means of the variables analyzed between nursing students and nurses.
|
|
Nursing students |
Nurses |
|
|
||
|
M |
SD |
M |
SD |
t
|
p |
|
|
PT |
4.52 |
0.45 |
4.52 |
0.57 |
-0.21 |
0.834 |
|
CC |
1.88 |
0.73 |
1.88 |
0.90 |
-0.01 |
0.991 |
|
TP |
2.35 |
1.03 |
2.05 |
1.04 |
5.25 |
0.000 |
|
EA |
3.63 |
0.71 |
3.58 |
0.77 |
1.08 |
0.028 |
|
EC |
3.54 |
0.75 |
3.84 |
0.69 |
-0.71 |
0.000 |
|
ER |
3.70 |
0.73 |
3.82 |
0.77 |
-3.01 |
0.003 |
|
Affective |
1.65 |
0.67 |
1.60 |
0.89 |
0.98 |
0.328 |
|
Cognitive |
4.73 |
0.63 |
4.49 |
0.82 |
5.53 |
0.000 |
|
Behavioral |
4.58 |
0.68 |
4.20 |
0.81 |
8.53 |
0.000 |
Note: EA= Emotional attention; EC = Emotional clarity; ER = Emotional repair;
PT= Perspective Taking; CC= Compassionate Care; TP= Thinking like the patient
3.3 Hierarchical regression model (HRM)
For the nursing students, two steps were established in the model for predicting the affective dimension of ACO (R2adjusted=0.06, p≤0.001). First, all the dimensions of JSE were entered (step 1) (R2 =0.54, p≤0.001). The TMMS24 components were then included (step 2) (R2 =0.01; p≤0.001). When each dimension was considered, only compassionate care (affective: β=0.09; p≤0.01) and thinking like the patient (affective: β=0.08; p≤0.05) were significant positive predictors of affective dimensions, and perspective-taking (affective: β=-0.15; p≤0.001) and emotional repair (affective: β=-0.08; p≤0.05) were significant negative predictors of affective dimensions (Table 2).
Two steps were established in the model for the prediction of cognitive dimension of ACO among nursing students (R2adjusted=0.18, p≤0.001). First, JSE components of empathy were entered (step 1) (R2 =0.17, p≤0.001). The all the trait components of EI were then included (step 2) (R2 =0.01, p≤0.01). The perspective-taking of empathy (cognitive: β=0.35; p≤0.001), was significant positive predictors, and compassionate care of empathy (cognitive: β=-0.16; p≤0.001), emotional attention(cognitive: β=-0.70; p≤0.05) and repair (cognitive: β=-0.70; p≤0.05), were significant negative predictors of cognitive dimension.
Three steps were established in the model for the prediction of the behavioral dimension of ACO (R2adjusted=0.73, p≤0.001). First, all the dimensions of the instrument JSE were entered (step 1) (R2 =0.15, p≤0.001). The TMMS24 dimensions were then included (step 2) (R2 =0.20, p≤0.001) and the affective and cognitive dimension were entered (step 3) (R2 =0.57, p≤0.001), according to the theoretical foundation. In the third step of prediction, the behavioral dimension, perspective-taking (β=0.04; p≤0.05) was a significant and positive predictor, while thinking like the patient of empathy (β =-0.05; p≤0.01) was a significant negative predictor. The emotional clarity of EI (β=0.05; p≤0.01) was a significant and positive predictor in the behavioral dimension. In the third step, attitudinal variables were more important than empathy or EI, with the best predictors being the cognitive dimension (β =0.80; p≤0.001) and the affective dimension (β =-0.09; p≤0.001).
EI and empathy account for 6% of the variance in affective dimension, 18% of the variance in cognitive dimension and 20% of the variance in behavioral dimensions. Empathy is a better predictor than EI in all cases, and especially the perspective-taking dimension in the cognitive and behavioral dimensions. In addition, in the prediction of behavioral dimension, the attitude components (affective and cognitive dimensions) (R2=0.57; p≤0.001) account for a greater degree of variance as emotional components. In general, empathy (perspective taking) and EI (emotional clarity and repair) predict ACO (affective and cognitive dimensions). These results for nurses were presented in Giménez-Espert and Prado-Gascó [49]”.
We appreciate your concern, about the discussion section. The authors have tried to clarify this section and include relevant literature to improve the discussion of the results, taking into account that there are no previous studies comparing these variables in both types of samples (nurses and nursing students).
Page 9-11, line 311-411
“Discussion
This research analyzed the differences between nursing students and nurses for the means of the variables measured (empathy, EI and attitudes towards communication), as well as the impact of empathy and EI on nursing students' attitudes towards patient communication and their influence on behavioral. This study is the first in nursing students that attempts to identify the relationships between emotional and attitudes towards communication. Therefore, the results obtained may be useful for improving nursing education and the quality of nursing care [23, 35,36].
In general, as was expected and postulated in the first hypothesis, statistically significant differences were observed for almost all the variables considered (with the exception of perspective taking, compassionate care and the affective dimension). In general, higher levels were observed for the students than the nurses, except in the case of dimension thinking like the patient of empathy written in negative, and emotional clarity and repair, where this situation is reversed. These results may be explained by the fact that as clinical experience in real and complex contexts increases, there is a decline in empathy in nursing students [64]. High levels of emotional attention with low levels of emotional clarity and repair are related to inadequate coping strategies (more passive and less active), psychological problems [65, 66], as an adaptive response to new responsibilities and an increasing workload [23]. Attention is focused on tasks and technology, to the detriment of relations with the patient [67]. Nursing students distance themselves from patient´s suffering as their clinical experience increases [4,68], which leads to lower levels of these emotional variables affecting patient safety [20]. This is because their training has been more focused on clinical skills, and they feel poorly trained to deal with these situations [21,22].
According to the second hypothesis, empathy ad EI will have a negative effect on the affective dimension and a positive effect on the cognitive and behavior dimension of attitudes towards communication in nursing students and nurses. The HRM results suggested that hypothesis H2 can be partially accepted. The relationships observed were in the direction hypothesized, although not all the dimensions were equally significant. Empathy and EI have a negative effect on the affective dimension and a positive effect on the cognitive and behavior dimension of attitudes towards communication in nursing students and nurses. Only the perspective-taking in empathy and the emotional repair dimension of the EI were significant negative predictors in the affective dimension, and compassionate care and thinking like the patient in empathy, was a significant positive predictor for the affective dimension in the nursing students. Meanwhile, in the nurses, the perspective-taking in empathy and emotional clarity dimension of EI were significant negative predictors in the affective dimension of attitudes towards communication. The perspective-taking in empathy was significant positive predictor of cognitive dimension in the nursing students, and in the nurses. The compassionate care of empathy and emotional attention and repair of EI were significant negative predictors of cognitive dimension in nursing students. The emotional clarity was significant positive predictor of cognitive dimension in nurses. In the behavioral dimension, perspective-taking was a significant and positive predictor, while thinking like the patient of empathy was a significant negative predictor, in the nursing students and nurses. In the nurses, emotional repair and in nursing students emotional clarity were significant and positive predictors. In addition, attitudinal variables were more important than empathy or EI in the nursing students and nurses, with the best predictors being the cognitive dimension (a significant positive predictor) and the affective dimension (a significant negative predictor).
Likewise, it seems that emotional variables were in general better predictors for the nurses than for the students, especially the perspective taking dimension of empathy in the cognitive, affective and behavioral dimensions in both samples of nursing students and nurses. In general, empathy (perspective taking) and EI (emotional clarity in nurses, and emotional repair in the nursing students) predicted their attitude towards communication (affective and cognitive dimensions). This could be because the perspective-taking dimension of empathy is the central cognitive axis of empathy, emotional clarity and emotional repair (high levels indicate adequate emotional adjustment) promoting favorable attitudes towards communication in order to better understand the patients [69,70]. In the behavioral dimension of attitude towards communication, the emotional component (EI and empathy) is a weaker predictor than the affective and cognitive dimensions, and the inclusion of the attitudinal component (affective and cognitive) significantly increased prediction to 57% of variance in the behavioral dimension of the attitude [49]. These findings can be explained by the fact that the three components of attitude are related, so that feelings (the affective dimension) are based on knowledge, in this case about communication (the cognitive dimension), and communicative behavior (the behavioral dimension) is based on feelings and knowledge about communication [21,71]. The cognitive and affective dimensions were better predictors of the behavioral dimension, as shown in the literature [13,28]. Moreover, empathy may be a predictor of EI [70,38]. High levels of empathy provide an understanding of internal feelings and those of others, distinguishing emotions as well as enabling listening [72], and more harmonious interpersonal relationships [73, 74], which are key elements of EI [71]. In fact, some authors include empathy as a part of EI [38, 75, 76], so according to the results, in nursing it seems more advisable to focus first on forming the cognitive and affective aspects of attitudes towards communication through training in emotional competency, especially empathy, for both nurses and nursing students in order to promote the behavioral aspect of attitudes towards communication with patients [77].
Despite the interest of this study, one of the major limitations is its convenience sampling. It is difficult to generalize the results, and the cross-sectional nature of the study type meant it was impossible to establish causal relationships. In future research, it would be interesting to extend the study sample to other Spanish-speaking countries, and to establish a longitudinal design that would enable causal relationships to be established, and to include other variables that may influence the results, such as age and gender [36]. Another limitation is related to the use of self-reporting, which can introduce social-desirability bias [62]. It would be useful to use another type of instrument completed by others, and/or one with external objective measures.
As for the practical implications of this research, given that EI and empathy are an integral part of nurse-patient communication, and can be learned and developed through education [78], these results could help managers in academic and healthcare institutions to highlight the importance of empathy in improving attitudes towards communication with patients by both nursing students and nurses to promote their well-being and provide high quality nursing care [23, 35,36]. It is a key area of nursing education, as resolving a wide variety of complex situations with patients requires emotional skills and positive attitudes towards communication [79]. Including education about emotional skills and communication in the nursing curricula, and in continuing education strategies for nurses is therefore a necessity [80]. Role-playing exercises [81] as well as clinical simulation [43] could be useful. Nurses could simulate real situations to feel the patient's physical state and emotional experience, which could improve nurses' EI and empathy. Educators of nurses could assess EI, empathy, and attitudes towards communication among nursing students on admission to training in order to develop an individual support plan for improvement [16], as well as in nurses, paying more attention to psychological counseling in their work environment, especially the wake of the COVID19 pandemic, and through regular continuing education courses to improve these skills [38]”.
Please see the attachment.

Reviewer 3 Report
Comments to the authors
This is an interesting manuscript. The article needs changes. See other comments below.
Abstract
Keywords: include “cross-sectional”.
Introduction and Background:
1.- Please, searches must be up to date. Check the information sources of the entire article. There are some works that have been updated and show more current data.
2- I suggest including more recent research that examines the problem. Please give specific data on the study area. For example: (line 67: Training in these skills is therefore an important tool for effective communication). Include current model applied in Spanish Universities to train these skills. Please add these ideas that help the reader to contextualize the problem.
Materials and Methods
Include mail objective in first paragraph.
Methods
1. Design: Please define cross-sectional method using different sources.
2. You say: “convenience sample of 961 nursing students recruited in Spain; The comparison sample was a convenience sample of 460 nurses in Spain”. These sentences are very important and need to be detailed:
Define “convenience sample”. How was the sample calculation done?
Participants and sampling method: How was the study presented to the participants? Explain the process of how the participants knew about the existence of this study. More information is needed to explain how the study is presented to the participants: Was it in a face-to-face, by phone, by e-mail? How do the researches access to the personal data of the nurses/students? Were they (participants) able to read the informed consent carefully? Where and when did the nurse/students complete the questionnaire? Include approximate response time. Please, add this information and give more details about informant consent in this section.
Need to give more detail on inclusion/exclusion criteria. What type of emotional or physical health situations contra-indicated study participation? Please explain this process in detail.
3. The data collection from nurses was conducted in September 2015–February 2016. When data were collected from students? This important information is not added. Please explain.
In relation to this issue, many important events have taken place in the healthcare environment since 2016... COVID pandemic has greatly influenced emotional management and communication. This has not been mentioned... Please, this information should be explained and adjusted to the current time. As I previously indicated, the search for information must be updated to 2023.
4. Type of sampling. Cite source.
5. Ethical consideration. Include the approval code in this section.
Results
Well done!
Discussion
The discussion does not correspond to the extent of the results or the topics covered in it. The discussion is much more synthetic than the results obtained. Please check it.
Some reorganization is required. The discussion is correct but somewhat disordered. I suggest writing the discussion in more order, giving the information according to the stated objectives and the mail results.
Include a section on Implications for Research/practice (with a subheading).
Strengths and limitations
Very honest section.
Conclusion
The conclusions are focused on the implications for practice. I suggest focusing the conclusion on the main ideas that the study brings to the world. What does your study contribute in the clinical context and for the nursing discipline?
Author Response
Dear reviewer,
We would like to thank you for your time and effort in revising our paper. We really appreciate the suggestions made and we believe the document has greatly benefited from it.
In general, abstract, introduction, materials and methods, results, discussion (limitations) and conclusions have been modified. More references have been included to cover the relevant literature. References have been reviewed, and some of them have been adapted in order to comply with the journal´s requirements. The article has undergone a new language revision, certificate of which is attached. All changes have been included in the article in red.
Response to reviewer 3
We appreciate your concern about the keywords. The authors have included “cross-sectional”.
Page 1, line 30-31
Keywords: attitude; communication; cross-sectional; empathy; emotional intelligence; nurses; nursing students.
We really appreciate the suggestions made about the introduction and background section. The authors have included some updated references and one current model applied in Spanish Universities to train these skills to contextualize the problem.
Page 1-3 line 33 -106
“1. Introduction
Nurse-patient communication is interpersonal communication aimed at assessing the patient’s real needs and thereby establishing a therapeutic relationship with him or her [1]. Therapeutic communication is a process of information exchange, an essential competence that nurses need to acquire to understand patients' needs, in order to plan appropriate care [2]. It is considered one of the most important communication methods and a central element of nursing care for the quality of care and patient satisfaction [3-5]. Nurses' ability to communicate effectively has been associated to patient engagement, and improved health outcomes [6]. In multidisciplinary teams, it is important for healthcare teamwork [7] and fewer adverse patient events [8].
In this context, and for appropriate, safe and quality nursing care [4,5], studying nursing students' attitudes, in this case towards patient communication, is vital for understanding beliefs and behaviors, and is a key factor in determining what is relevant in the process of care and healing [9]. Furthermore, the assessment of nursing students' attitudes towards patient communication is therefore critical for a better understanding of behavior and to ensure adequate communicative competence [1]. The "Theory of Reasoned Action" [10] shows the relationship between the attitudes and behavior of individuals. According to this theory, a change in a person's attitude can motivate a change in his or her behavior, and as such studying attitudes determines the probability of an individual performing the behavior in question [11]. Attitude towards communication is made up of three dimensions: cognitive (beliefs and thoughts about communication), affective (positive or negative emotions of a person towards communication) and behavioral (the behavioral disposition, which is a manifestation of the cognitive and affective dimensions) [12]. This aspect is very important for nursing students and nurses, because the assessment of attitudes towards patient communication identifies negative attitudes that can influence the effectiveness of the educational process [13] and the quality of nursing care [14]. Developing favorable attitudes towards communication in nursing students is therefore critical [15], and communication is important in improving patient care [16].
Additionally, the literature suggests the existence of personal variables that influence communication with the patient, such as emotional intelligence (EI) and empathy [17,18]. Nurses must understand the needs of their patients, i.e. they must be able to adequately organize their emotions and those of others (EI) and above all, be able to put themselves in the other's place (empathy) [19], as fundamental aspects for effective communication. Patient safety and nurses' well-being can be jeopardized if the use of these skills is poorly [20]. Nursing education should also teach communication skills, EI and empathy for adequate patient attention [21,22]. These skills are necessary for safe nursing care, teamwork, health care and management skills essential for decision making and problem solving [23,24]. These aspects are also critical for the delivery of safe and quality nursing care to patients [23]. Communication skills, empathy and EI are fundamental aspects for identifying patients’ needs, increasing nurses' clinical competence and providing better patient care [25]. However, they are not evaluated during nursing students’ training [21]. Although the literature indicates that the levels of these variables may vary according to the participants’ characteristics and as the training of nursing students progresses, after being exposed to complex real-life situations and human suffering without adequate preparation and training [26-28], the explanation may lie in the time constraints of curricula which lead to nurses' clinical competence being prioritized rather than their emotional skills [29].
According to the literature, the most patient complaints in hospitals are caused by healthcare professionals’ negative attitudes or poor communication skills [30,31]. EI and empathy are an integral part of nurse-patient communication, and can be learned and developed through education [32]. These skills are interrelated in the development of nursing care empathy and EI are positively related to nurses' job satisfaction [33], students' mental health and academic performance [34], as well as professional attitude [31]. They also improve the effectiveness of nurses, reducing the incidence of adverse events and job burnout, enhance the nurse-patient relationship and the quality of nursing care overall [35,36]. Training in these skills is therefore an important tool for effective communication, EI, empathy and patient care [37], as well as for the well-being of nurses [38]. The adoption of a framework in universities with the fundamentals of nursing care focused on the essential needs of the person to ensure their physical and psychosocial well-being is therefore essential in the education of nursing students [39]. These needs are met by developing a positive and trusting relationship with the person being cared for, as well as with his or her family [40]. A positive and trusting nurse-patient relationship based on effective communication is essential for providing high-quality basic care [41]. Once this relationship is established, the nurse can work to meet the patient's fundamental psychosocial and physical needs. Meeting these needs is in turn mediated by nurses' emotional skills such as communication, empathy, and EI [38,42,43]. Another key element is the care context, including health policies that may hinder or facilitate high-quality basic care [44]. In addition, the importance of the global nursing workforce in the sustainability of health systems has been demonstrated in the context of the COVID-19 pandemic [45]. The training of nurses and nursing students as future professionals in these aspects is therefore essential for improving their well-being and job satisfaction [38]. These are crucial issues, since 43% of nurses in Europe consider leaving their job within three years [46]”.
We appreciate your concern about the objectives. The authors, in accordance with the reviewer's suggestion, have rewritten the objectives in clearer way and write the hypothesis.
Page 3 line 114-126
“The main aims of this study are therefore (1) to explore the differences between nursing students and nurses in the means of the variables analyzed (empathy, EI and attitudes towards communication); and (2) to assess the relationship between empathy and emotional intelligence (EI) as predictors of nursing students' attitudes towards communication, and their influence on communication-related behavior using regression, making a comparison with the results in nurses [49]. On the basis of the previous literature, the following hypotheses delimited this study:
- H1: All analyzed variables (empathy, EI and attitudes towards communication) will show statistically significant differences in their means in nursing students and nurses.
- H2: Empathy and EI will have a negative effect on the affective dimension and a positive effect on the cognitive and behavior dimension of attitudes towards communication in nursing students and nurses.”
We really appreciate the suggestions made about the methods. The authors have defined cross-sectional method using different sources (page 3 line 138-141), “convenience sample” (page 3-4, line 141-144) and how the calculation size was done (nursing students page 3, line 134-138; nurses page 4, line 157-161).
Page 3 line 138-141
“A cross-sectional study is a type of observational study design that involves analyzing data from a population at a specific point in time, with no prospective or retrospective follow-up. Subjects are selected from a population available for the study question, showing the "point in time" situation of a sample of subjects [54,55]”.
Page 3-4, line 141-144
“Convenience sampling is a non-probability sampling method, in which the sample is selected according to a subjective judgment of the researcher, based on the subjects' availability and willingness to participate [55,56]”.
Nursing students page 3, line 134-138
“A cross-sectional descriptive study was conducted in a convenience sample of 961 nursing students, recruited in five universities in the Valencian Community, Spain. The sample size was determined according to data available from the Spanish Ministry of Universities, of 5,062 nursing students enrolled in the Valencian Community in the 2018/2019 academic year, with a margin of error of 2.85% and a confidence level of 95% [53]”.
Nurses page 4, line 157-161
“The comparison sample was a convenience sample of 460 nurses enrolled from 6 hospitals in the Valencian Community, Spain [49]. The sample size was determined according to data available from the Spanish National Statistics Institute, with a total of 26,565 nurses working in the Valencian Community in 2015, with a margin of error of 4.53% and a confidence level of 95% [53]”.
In addition, we have included more information about the participants, sampling method and data collection (nursing students, page 4, line 147-156; nurses page 4, line 161-171), inclusion/exclusion criteria (page 4, line 144-147), type of sampling with source (page 3-4, line 141-144), and the ethical approval code (page 4, line 172-174), according to the reviewer suggestions.
Nursing students page 4, line 147-156
“The data were collected by researchers at the selected universities in the 2018/2019 academic year, using an instrument that included sociodemographic variables (such as sex, age, year of degree course and employment situation of the respondents), attitudes towards communication (ACO) adapted to nursing students [28], the Trait Meta Mood Scale (TMMS24) [57] and the Jefferson Scale of Empathy (JSE) for nursing students adapted and validated in Spain by Giménez – Espert et al. [57]. Approximately 20 minutes were required to complete the instrument, and it was completed in the classroom by the students. All the participants were informed of the aims of the study, the confidentiality of the data, and that non‐participation would not have any consequences for their academic development”.
Nurses page 4, line 161-171
“The data collection from the nurses was conducted by researchers in September 2015–February 2016. The instrument contained sociodemographic variables, attitudes towards communication of nurses (ACO) [58], Trait Meta-Mood Scale (TMMS24) [59], and Jefferson Scale Nursing Empathy (JSNE) [60,61]. Approximately 35 minutes were required to complete the instrument. The researchers visited the hospitals involved in the study. All the participants received detailed information about the aims and procedures, and were informed regarding confidentiality, with an emphasis on the anonymity of the data collected and non-discrimination of participants. Once the instruments had been submitted, reminders were sent via e-mail after 2 weeks and after 3-4 weeks. The completed instruments were subsequently collected from the various boxes placed in the different services for this purpose [49]”.
Page 4, line 144-147
“The inclusion criteria were (i) voluntarily agreeing to participate and (ii) signing the informed consent. Exclusion criteria were not having been diagnosed with a severe physical or psychiatric illness preventing the student’s presence during the application of the instrument”.
Page 3-4, line 141-144
“Convenience sampling is a non-probability sampling method, in which the sample is selected according to a subjective judgment of the researcher, based on the subjects' availability and willingness to participate [55,56]”.
Page 4, line 172-174
This study complied with the basic principles of the Helsinki Declaration (World Medical Association, 2013). It was authorized by the Human Research Ethics Committee of the University of Valencia (H1529396558647).
We appreciate your concern about the need of updated references and the influence of the COVID19 pandemic (page 3 line 101-106, page 10-11, line 405-411). The authors have included more updated references (9 of them from the last 3 years, reference number 24, 25, 36, 38, 39, 41, 43, 45 and 53).
Page 3 line 101-106
In addition, the importance of the global nursing workforce in the sustainability of health systems has been demonstrated in the context of the COVID19 pandemic [45]. The training of nurses and nursing students as future professionals in these aspects is therefore essential for improving their well-being and job satisfaction [38]. These are crucial issues, since 43% of nurses in Europe consider leaving their job within three years [46].
Page 10-11, line 405-411
“Nurses could simulate real situations to feel the patient's physical state and emotional experience, which could improve nurses' EI and empathy. Educators of nurses could assess EI, empathy, and attitudes towards communication among nursing students on admission to training in order to develop an individual support plan for improvement [16], as well as in nurses, paying more attention to psychological counseling in their work environment, especially the wake of the COVID19 pandemic, and through regular continuing education courses to improve these skills [38]”
We really appreciate the suggestions made about discussion section. The authors have reorganized and ordered this section to clarify the information according to the stated objectives and the main results. In addition, the authors have included the implications for research/ practice (page 10-11, line 395-411), strengths and limitations (page 10 line 380-394)according with the reviewer's suggestions.
Page 9 -11, line 311-411
- 4. Discussion
This research analyzed the differences between nursing students and nurses for the means of the variables measured (empathy, EI and attitudes towards communication), as well as the impact of empathy and EI on nursing students' attitudes towards patient communication and their influence on behavioral. This study is the first in nursing students that attempts to identify the relationships between emotional and attitudes towards communication. Therefore, the results obtained may be useful for improving nursing education and the quality of nursing care [23, 35,36].
In general, as was expected and postulated in the first hypothesis, statistically significant differences were observed for almost all the variables considered (with the exception of perspective taking, compassionate care and the affective dimension). In general, higher levels were observed for the students than the nurses, except in the case of dimension thinking like the patient of empathy written in negative, and emotional clarity and repair, where this situation is reversed. These results may be explained by the fact that as clinical experience in real and complex contexts increases, there is a decline in empathy in nursing students [64]. High levels of emotional attention with low levels of emotional clarity and repair are related to inadequate coping strategies (more passive and less active), psychological problems [65, 66], as an adaptive response to new responsibilities and an increasing workload [23]. Attention is focused on tasks and technology, to the detriment of relations with the patient [67]. Nursing students distance themselves from patient´s suffering as their clinical experience increases [4,68], which leads to lower levels of these emotional variables affecting patient safety [20]. This is because their training has been more focused on clinical skills, and they feel poorly trained to deal with these situations [21,22].
According to the second hypothesis, empathy ad EI will have a negative effect on the affective dimension and a positive effect on the cognitive and behavior dimension of attitudes towards communication in nursing students and nurses. The HRM results suggested that hypothesis H2 can be partially accepted. The relationships observed were in the direction hypothesized, although not all the dimensions were equally significant. Empathy and EI have a negative effect on the affective dimension and a positive effect on the cognitive and behavior dimension of attitudes towards communication in nursing students and nurses. Only the perspective-taking in empathy and the emotional repair dimension of the EI were significant negative predictors in the affective dimension, and compassionate care and thinking like the patient in empathy, was a significant positive predictor for the affective dimension in the nursing students. Meanwhile, in the nurses, the perspective-taking in empathy and emotional clarity dimension of EI were significant negative predictors in the affective dimension of attitudes towards communication. The perspective-taking in empathy was significant positive predictor of cognitive dimension in the nursing students, and in the nurses. The compassionate care of empathy and emotional attention and repair of EI were significant negative predictors of cognitive dimension in nursing students. The emotional clarity was significant positive predictor of cognitive dimension in nurses. In the behavioral dimension, perspective-taking was a significant and positive predictor, while thinking like the patient of empathy was a significant negative predictor, in the nursing students and nurses. In the nurses, emotional repair and in nursing students emotional clarity were significant and positive predictors. In addition, attitudinal variables were more important than empathy or EI in the nursing students and nurses, with the best predictors being the cognitive dimension (a significant positive predictor) and the affective dimension (a significant negative predictor).
Likewise, it seems that emotional variables were in general better predictors for the nurses than for the students, especially the perspective taking dimension of empathy in the cognitive, affective and behavioral dimensions in both samples of nursing students and nurses. In general, empathy (perspective taking) and EI (emotional clarity in nurses, and emotional repair in the nursing students) predicted their attitude towards communication (affective and cognitive dimensions). This could be because the perspective-taking dimension of empathy is the central cognitive axis of empathy, emotional clarity and emotional repair (high levels indicate adequate emotional adjustment) promoting favorable attitudes towards communication in order to better understand the patients [69,70]. In the behavioral dimension of attitude towards communication, the emotional component (EI and empathy) is a weaker predictor than the affective and cognitive dimensions, and the inclusion of the attitudinal component (affective and cognitive) significantly increased prediction to 57% of variance in the behavioral dimension of the attitude [49]. These findings can be explained by the fact that the three components of attitude are related, so that feelings (the affective dimension) are based on knowledge, in this case about communication (the cognitive dimension), and communicative behavior (the behavioral dimension) is based on feelings and knowledge about communication [21,71]. The cognitive and affective dimensions were better predictors of the behavioral dimension, as shown in the literature [13,28]. Moreover, empathy may be a predictor of EI [70,38]. High levels of empathy provide an understanding of internal feelings and those of others, distinguishing emotions as well as enabling listening [72], and more harmonious interpersonal relationships [73, 74], which are key elements of EI [71]. In fact, some authors include empathy as a part of EI [38, 75, 76], so according to the results, in nursing it seems more advisable to focus first on forming the cognitive and affective aspects of attitudes towards communication through training in emotional competency, especially empathy, for both nurses and nursing students in order to promote the behavioral aspect of attitudes towards communication with patients [77].
Despite the interest of this study, one of the major limitations is its convenience sampling. It is difficult to generalize the results, and the cross-sectional nature of the study type meant it was impossible to establish causal relationships. In future research, it would be interesting to extend the study sample to other Spanish-speaking countries, and to establish a longitudinal design that would enable causal relationships to be established, and to include other variables that may influence the results, such as age and gender [36]. Another limitation is related to the use of self-reporting, which can introduce social-desirability bias [62]. It would be useful to use another type of instrument completed by others, and/or one with external objective measures.
As for the practical implications of this research, given that EI and empathy are an integral part of nurse-patient communication, and can be learned and developed through education [78], these results could help managers in academic and healthcare institutions to highlight the importance of empathy in improving attitudes towards communication with patients by both nursing students and nurses to promote their well-being and provide high quality nursing care [23, 35,36]. It is a key area of nursing education, as resolving a wide variety of complex situations with patients requires emotional skills and positive attitudes towards communication [79]. Including education about emotional skills and communication in the nursing curricula, and in continuing education strategies for nurses is therefore a necessity [80]. Role-playing exercises [81] as well as clinical simulation [43] could be useful. Nurses could simulate real situations to feel the patient's physical state and emotional experience, which could improve nurses' EI and empathy. Educators of nurses could assess EI, empathy, and attitudes towards communication among nursing students on admission to training in order to develop an individual support plan for improvement [16], as well as in nurses, paying more attention to psychological counseling in their work environment, especially the wake of the COVID19 pandemic, and through regular continuing education courses to improve these skills [38].
Practical implications for this research
Page 10-11, line 395-411
As for the practical implications of this research, given that EI and empathy are an integral part of nurse-patient communication, and can be learned and developed through education [78], these results could help managers in academic and healthcare institutions to highlight the importance of empathy in improving attitudes towards communication with patients by both nursing students and nurses to promote their well-being and provide high quality nursing care [23, 35,36]. It is a key area of nursing education, as resolving a wide variety of complex situations with patients requires emotional skills and positive attitudes towards communication [79]. Including education about emotional skills and communication in the nursing curricula, and in continuing education strategies for nurses is therefore a necessity [80]. Role-playing exercises [81] as well as clinical simulation [43] could be useful. Nurses could simulate real situations to feel the patient's physical state and emotional experience, which could improve nurses' EI and empathy. Educators of nurses could assess EI, empathy, and attitudes towards communication among nursing students on admission to training in order to develop an individual support plan for improvement [16], as well as in nurses, paying more attention to psychological counseling in their work environment, especially the wake of the COVID19 pandemic, and through regular continuing education courses to improve these skills [38].
Strengths and limitations
Page 10 line 380-394
“In fact, some authors include empathy as a part of EI [38, 75, 76], so according to the results, in nursing it seems more advisable to focus first on forming the cognitive and affective aspects of attitudes towards communication through training in emotional competency, especially empathy, for both nurses and nursing students in order to promote the behavioral aspect of attitudes towards communication with patients [77].
Despite the interest of this study, one of the major limitations is its convenience sampling. It is difficult to generalize the results, and the cross-sectional nature of the study type meant it was impossible to establish causal relationships. In future research, it would be interesting to extend the study sample to other Spanish-speaking countries, and to establish a longitudinal design that would enable causal relationships to be established, and to include other variables that may influence the results, such as age and gender [36]. Another limitation is related to the use of self-reporting, which can introduce social-desirability bias [62]. It would be useful to use another type of instrument completed by others, and/or one with external objective measures”.
We appreciate your concern about the conclusion section. The authors have rewritten this section to focus on the contributions to the clinical context and nursing discipline.
Page 11, line 412-421
“5. Conclusions
This research can be considered an initial approach to the study of EI and empathy as predictors of nursing students' attitudes towards communication compared with a sample of nurses. Developing empathy and the cognitive dimension of attitudes towards communication in nursing students and nurses could help to improve EI and attitudes towards communication with patients and their families. The findings of the study produce scientific evidence for further investigation to identify and develop intervention programs adjusted to real needs based on individual support plans, aimed at improving the education of nursing students and involving regular continuing education courses for nurses in order to ensure their well-being and improve the quality of patient care”.
Please see the attachment

Round 2
Reviewer 3 Report
The paper has improved considerably. All suggestions have been incorporated. Congratulations to the research team.